# Ecological Civilisation and Amphibian Sustainability through Reproduction Biotechnologies, Biobanking, and Conservation Breeding Programs (RBCs)

**DOI:** 10.3390/ani14101455

**Published:** 2024-05-14

**Authors:** Robert K. Browne, Qinghua Luo, Pei Wang, Nabil Mansour, Svetlana A. Kaurova, Edith N. Gakhova, Natalia V. Shishova, Victor K. Uteshev, Ludmila I. Kramarova, Govindappa Venu, Somaye Vaissi, Zeynab Taheri-Khas, Pouria Heshmatzad, Mikhail F. Bagaturov, Peter Janzen, Renato E. Naranjo, Aleona Swegen, Julie Strand, Dale McGinnity, Ilze Dunce

**Affiliations:** 1Sustainability America, Sarteneja, Belize; 2School of Biological Resources and Environmental Sciences, Jishou University, Jishou 416000, China; dani2017@126.com (Q.L.); wangpei0229@126.com (P.W.); 3College of Biological and Chemical Engineering, Changsha University, Changsha 410022, China; 4Fujairah Research Centre (FRC), Al-Hilal Tower 3003, Fujairah P.O. Box 666, United Arab Emirates; nabil.mansour@frc.ae; 5Institute of Cell Biophysics, Russian Academy of Sciences, Pushchino 142290, Russia; sakaurova@mail.ru (S.A.K.); gakhova@gmail.com (E.N.G.); cryopreservation@list.ru (N.V.S.); uteshev-cryobank@mail.ru (V.K.U.); 6Institute of Theoretical and Experimental Biophysics, Russian Academy of Sciences, Pushchino 142290, Russia; luda_kramarova@rambler.ru; 7Centre for Applied Genetics, Department of Zoology, Jnana Bharathi Campus, Bangalore University, Bengaluru 560056, India; venugcaecilian@gmail.com; 8Evolving Phylo Lab, Centre for Ecological Sciences, Indian Institute of Science, Bengaluru 560012, India; 9Department of Biology, Faculty of Science, Razi University, Kermanshah 57146, Iran; s.vaissi@razi.ac.ir (S.V.); zeynab.taheri93@gmail.com (Z.T.-K.); 10Department of Fisheries, Faculty of Fisheries and Environmental Sciences, Gorgan University of Agricultural Sciences and Natural Resources, Gorgan 49138, Iran; pheshmatzad@gmail.com; 11IUCN/SSC/Athens Institute for Education and Research/Zoological Institute RAS, St. Petersburg 199034, Russia; bbigmojo@mail.ru; 12Verband Deutscher Zoodirectoren/Justus-von-Liebig-Schule, 47166 Duisburg, Germany; pjanzen@gmx.de; 13Centro Jambatu de Investigación y Conservación de Anfibios, Fundación Jambatu, Giovanni, Farina 566 y Baltra, San Rafael, Quito 171102, Ecuador; renatonaranjo53@gmail.com; 14School of Environmental and Life Sciences, College of Engineering, Science and Environment, University of Newcastle, Callaghan 2308, Australia; aleona.swegen@newcastle.edu.au; 15Department of Animal and Veterinary Science, Aarhus University, Blichers Alle 20, 8830 Tjele, Denmark; js@biosfaeren.dk; 16Ectotherm Department, Nashville Zoo at Grassmere, Nashville, TN 37211, USA; dmcginnity@nashvillezoo.org; 17Amphibia Lab, 1307 Riga, Latvia; ilzedunce@gmail.com

**Keywords:** COP 15, COP 28, biobanking, amphibian, bioregionalism, intergenerational justice, ART, multilateralism, de-extinction, effective altruism

## Abstract

**Simple Summary:**

Intergenerational justice entitles future generations to the maximum retention of Earth’s biodiversity. The 2022 United Nations COP 15, “Ecological Civilisation: Building a Shared Future for All Life on Earth”, aims to safeguard 30% of Earth’s terrestrial environment by 2030, and COP 28 addressed the climate catastrophe. Reproduction biotechnologies, biobanks, and conservation breeding programs (RBCs) are also needed to perpetuate amphibian diversity and prevent extinctions. We focused this review on three core themes: the need and potential of RBCs to satisfy sustainability goals, the technical state and current application of RBCs, and how to achieve the future potentials of RBCs in a rapidly evolving environmental and cultural landscape. The full potential of amphibian RBCs requires a democratic, globally inclusive organisation that focuses on developing facilities in the regions with the highest amphibian diversity.

**Abstract:**

Intergenerational justice entitles the maximum retention of Earth’s biodiversity. The 2022 United Nations COP 15, “Ecological Civilisation: Building a Shared Future for All Life on Earth”, is committed to protecting 30% of Earth’s terrestrial environments and, through COP 28, to mitigate the effects of the climate catastrophe on the biosphere. We focused this review on three core themes: the need and potential of reproduction biotechnologies, biobanks, and conservation breeding programs (RBCs) to satisfy sustainability goals; the technical state and current application of RBCs; and how to achieve the future potentials of RBCs in a rapidly evolving environmental and cultural landscape. RBCs include the hormonal stimulation of reproduction, the collection and storage of sperm and oocytes, and artificial fertilisation. Emerging technologies promise the perpetuation of species solely from biobanked biomaterials stored for perpetuity. Despite significant global declines and extinctions of amphibians, and predictions of a disastrous future for most biodiversity, practical support for amphibian RBCs remains limited mainly to a few limited projects in wealthy Western countries. We discuss the potential of amphibian RBCs to perpetuate amphibian diversity and prevent extinctions within multipolar geopolitical, cultural, and economic frameworks. We argue that a democratic, globally inclusive organisation is needed to focus RBCs on regions with the highest amphibian diversity. Prioritisation should include regional and international collaborations, community engagement, and support for RBC facilities ranging from zoos and other institutions to those of private carers. We tabulate a standard terminology for field programs associated with RBCs for publication and media consistency.

## 1. Introduction

Intergenerational justice entitles the maximum retention of Earth’s biodiversity [1]. This review focuses on three core themes: the need and potential of reproduction biotechnologies, biobanks, and conservation breeding programs (RBCs) to satisfy sustainability goals; the technical state and current application of RBCs; and how to achieve the future potentials of RBCs in a rapidly evolving environmental and cultural landscape. There are increasing rates of decline and extinction in amphibian species due to widespread anthropogenic damage to global ecosystems [2,3,4]. This crisis demands proactive and innovative strategies to perpetuate amphibian diversity. The 2022 United Nations COP 15, “Ecological Civilisation: Building a Shared Future for All Life on Earth”, through the protection of 30% of Earth’s terrestrial area [5], and COP 28, in mitigating the effects of the climate crisis [6], focus on sustainably managing the biosphere. Unfortunately, the climate crisis alone [3] will inevitably result in profoundly modifying ecosystems over the coming decades, leading to a disastrous cascade of species population declines and extinctions [3,4]. Consequently, a rapid increase in the number of amphibian species reaching extinction in the wild can be expected [2,3,4].

The development and application of RBCs can achieve the biodiversity conservation goals of COP 15 and COP 28. RBCs can safely, securely, and economically maintain endangered and critically endangered species and proliferate genetically varied individuals for release in field conservation programs [7,8,9,10,11,12,13,14,15]. However, the full potential of RBCs lies in perpetuating species through the restoration of individuals solely from biobanked cells or tissues [16,17,18,19,20,21]. RBCs’ economical and efficient use [13,14] is particularly valuable for species whose natural habitats will be lost and otherwise neglected [22,23,24].

The global establishment of amphibian RBCs requires developing networks to support facilities, particularly in the Global South and other countries with the highest amphibian diversity [25,26,27,28,29,30,31]. Two fundamental principles form the foundation of non-partisan multi-stakeholder engagement: firstly, the increasing assertion of the Global South and other developing countries in a multipolar geopolitical landscape [5,28], and secondly, the growing significance of non-governmental organisations (NGOs) and the public in species conservation [25]. We use the term developing countries with the caveat that our use of the term refers to modern eco-friendly development to provide human well-being and environmental sustainability rather than GDP growth as the critical indicator of development [5,28]. Following these principles can build RBCs through collaborations with regional, national, and international programs, an approach that enhances the global effort to protect amphibian diversity and contributes to the stewardship of local communities in RBC conservation initiatives [6,26,27,28,29,30,31,32,33,34].

Amphibians provide an optimal vertebrate class to implement and exemplify species management through RBCs. Cultural and political advantages of amphibian RBCs include amphibians’ popularity as conservation ambassadors and amenability to regional community programs, including ecosystem protection [26,27,28,29,30,31,32,33,34]. Biological, technical, and economic factors include amphibians’ small size, relative ease of care compared to megafauna, high fecundity, and proven amenability for RBCs [7,8,9,10,11,12,13,14,15]. This amenability fits well with the One Plan Approach to Conservation [35], which integrates the management of a species’ metapopulation [36] through in situ with ex situ approaches, including RBCs, to maintain or restore genetic variation [17,18,19,20,21,26,27,28,29,30,31,32,33,34].

Since the turn of the millennium, RBCs have perpetuated amphibian species’ genetic diversity and allelic variation [7,8,9,10,11,12,13,14,15]. This ability has led to well-financed programs in Australia [37] and the USA [38], with nascent programs in Global South countries with very high amphibian diversity, including Ecuador [39], Mexico [40], Panama [12], and Papua New Guinea [pers comm] lacking sufficient financial support. Consequently, the demonstrated capability of amphibian RBCs to reduce costs and increase the reliability of *global* ex-situ and in-situ species management still needs to be fulfilled [2,7,8,9,10,11,12,13,14,15,22,41,42].

We explore these potentials through the sections (1) Regional targeting of amphibian RBCs, (2) Species prioritisation for amphibian RBCs, (3) Maintaining genetic diversity, (4) Contextualising amphibian RBCs, (5) Amphibian conservation breeding programs (CBPs), (6) Biobanking facilities, (7) Financial support, (8) Cultural engagement, and (9) The road ahead, where we argue that the full potential of amphibian RBCs requires a democratic, globally inclusive organisation that focuses on developing facilities in the regions with the highest amphibian diversity.

## 2. Regional Targeting of Amphibian RBCs

Regional targeting of amphibian RBCs depends on biogeographical species richness and threats, including habitat loss and global heating, prey loss and increased predation, illegal trade, and pathogens and parasites.

The highest amphibian diversities are found in regions with long geological periods, amenable climates, and varied geomorphology, including the isolating mechanisms of islands, mountains, and watersheds to promote speciation [43,44]. In 2023, the AmphibiaWeb database catalogued 8713 amphibian species. Of these, 7677 are frogs and toads, 815 are newts and salamanders, and 221 are caecilians. The ten top countries in declining order of amphibian species richness were Brazil (1176 sp.) Colombia (832 sp.), Ecuador (688 sp.), Peru (672 sp.), China (607 sp.), India (454 sp.), Papua New Guinea (426 sp.), Mexico (424 sp.), Madagascar (412 sp.), Indonesia (394 sp.), and Venezuela (365 sp.) [45].

The ranges of the three orders of amphibians, anurans (frogs and toads), salamanders (Caudata), and caecilians (Gymnophiona) are not sympatric. Major biogeographical areas with high anuran diversity include Southeast Asia, Africa, Southeast North America, and Central and South America [43,44,45], with most new species discoveries in the tropical regions of South America, India/Asia, Indo-Pacific, and Africa [44]. Salamanders and caecilians are more restricted in distribution than anurans. Among the ten salamander families, nine are predominately temperate, with a centre of diversity in Southeast North America. However, species of the tenth family, the highly threatened Plethodontidae, are mainly tropical and include 228 of the 555 currently described salamander species [46,47,48,49]. Half of these species are in the Mesoamerican Highlands of Mexico or South and Central America, with many new species being discovered in Brazil [49]. The distribution of the basal salamander family Cryptobranchoidea extends from Northeast Asia to Eastern North America [50]. Caecilians inhabit Central and South America, the South Asian tropics, and Eastern and Western Africa, with an intriguing absence in Central Africa. The conservation status of many caecilian species is unknown due to their fossorial habitats [47,48,51].

Habitat loss is the primary driver of amphibian declines, affecting ~65% of all amphibian species and reaching alarming levels of ~90% for threatened species [52,53]. Targeting only 2% of the global terrestrial area would protect over 80% of salamander and 65% of caecilian and anuran phylogenetic diversity [54]. Much amphibian species diversity also exists in a small percentage of bioregions. For example, the small country of Ecuador is home to 8% of amphibian species, half of which are endemic [55]. Island regions with high anuran endemicity include Melanesia, with 15% of species in 0.7% of the global terrestrial area [56], and Madagascar, with 5% of species in 0.4% of the global terrestrial area [57]. Many amphibian species also depend on mountain habitats with a high altitudinal range [58,59], and many of these species will become extinct in the wild due to unachievable needs for altitudinal migration due to global heating to a likely 2.5 °C or more [3,4,60,61,62].

A significant threat to amphibians is reduced prey availability, where insect populations and their diversity are rapidly declining [63,64]. Modifications to aquatic ecosystems in flow and water quality can produce changes in the density and composition of biota that could affect tadpole growth and survival [65]. Exotic species are an increasing driver of species declines and extinctions [66], where complicated interactions can occur between species within ecosystems [67]. For instance, the invasive Cane toad *Rhinella marina* significantly affects some Australian native frog populations via rarefication through direct predation and competition, with counteracting of these effects by *R. marina* toxicity to frog populations in general [68].

Commercialising natural biological resources can yield conservation benefits through increased cultural engagement and habitat management or harm through over-exploitation. The commercial amphibian trade for food or companion animals includes thousands of individuals and hundreds of species [69,70]. To our knowledge, no amphibian species have become extinct through overcollection for trade. However, assessing the sustainability of this trade is challenging due to difficulties in tracking [70], exacerbated by insufficient taxonomic and threatened species data [71]. Unfortunately, CITES regulations are not sympathetic toward the extraordinary need of private carers for the international trade of listed species to support their CBPs. Meanwhile, criminal and irresponsible individuals and organisations profit through illegal trade [69].

Significant threats to amphibian survival are pathogens, including *Batrachochytrium dendrobatidis* (Bd), which spread globally in the early to mid-20th century, causing the extinction of ~2% of amphibian species and infecting ~7% of species, as was found in ~70% of surveyed countries [72,73]. Inter-regional transmission of *B. dendrobatidis* can occur through keratinous skin on crayfish, nematode worms [73], fish [74,75], and migratory birds’ feet [76]. Emerging threats from pathogens include *B. salamandrivorans*, confined to Eurasia but threatening the high salamander diversity of the Americas [77], and through global heating, which increases the ranges of amphibian parasites [50,78,79]. Safeguarding amphibians from emerging pathogens relies on factoring in pathogens, sound management, and control rather than elimination [80,81,82].

## 3. Species Prioritisation for Amphibian RBCs

Significant sources for species prioritisation for amphibian RBCs through endangerment status are the Amphibia Web [45], IUCN Red List [2], and the Amphibian Ark [42]. A different perspective is through The Royal Zoological Society of London, Evolutionarily Distinct and Globally Endangered (EDGE), which prioritises species based on phylogenetic significance [83,84]. RBCs can also be prioritised for amphibians displayed in zoos [85], in the general community as companion animals that exhibit attractive colours or patterns [25], or for species of cultural significance to traditional communities [25,86].

The term “Threatened Species” is often used generically concerning RBCs [87]. However, the Threatened Species category includes Vulnerable (VU), Endangered (EN), and Critically Endangered (CE) species [2]. Vulnerable species are not thematic targets of RBCs, as their management focus should be field conservation programs, including habitat protection or replacement, repopulation, translocation, and augmentation (Table 1, [33,35,88,89]).

Database integration between all components of RBCs is essential to provide contemporary and accurate information about targeted species [90,91,92]. Over half of the amphibian species on the IUCN Red List have outdated assessments [2], with this deficiency particularly applying to regions with high amphibian diversity [2,47,49] and caecilians [48,51]. Besides this deficiency, the IUCN Red List does not offer a functional list of species recommended for CBPs, termed “Ex-situ Conservation” [Table 1]. Instead, species recommended for CBPs are mostly Least Concern (LC) species.

A framework for evaluating the relevance and impact of the IUCN Red List of Threatened Species was published in 2020 [93] but has yet to be implemented. Unfortunately, the annual funding of the IUCN Red List and other significant databases for amphibian sustainability is inadequate and only represents a minuscule amount of global biospheric sustainability funding. Adequate financial support for these databases and associated taxonomy would support amphibian sustainability and help save species in other taxa.

Palacio et al. [94] also showed that inaccuracies in the Aves Red List hampered rather than supported their conservation. Furthermore, they found that Red List specialist groups did not respond to feedback and needed more transparency, and the Red List rules and guidelines were either not followed or misused.

The Amphibian Ark was established through collaboration between the World Association of Zoos and Aquariums (WAZA), the IUCN SSC Conservation Planning Specialist Group (CPSG), and the IUCN SSC Amphibian Specialist Group (ASG [88]). The AArk provides contemporary information, a newsletter, seed grants for CBPs, and regional assessments for CBPs listing over 400 EN and CE species, with almost 100% recommended for RBCs [42]. However, the AArk and the IUCN prioritisation for RBCs need improvement to ensure the long-term sustainability of amphibian biodiversity. For instance, besides their endangerment status [2], Amphibian Ark prioritisation of species suitability for CBPs also includes other criteria [22]:Biodiversity loss is acceptable in principle. However, many consider that it is ethically unacceptable for amphibian species to be pre-emptively doomed to extinction due to neglect through policy [95]. Furthermore, in line with global environmental standards, the USA and Australian governments have adopted a no-species-lost policy [96].Candidate species for CBPs need an evidenced potential for eventual repopulation in the wild [97,98]. However, this mandate could result in the neglect of many species as ecosystems are predictably modified or destroyed. Furthermore, it is challenging to predict the future potential survival of species in the wild due to adaptation to pathogens [99], through amelioration [80,82], or via release programs for Bd-prone species [100,101].The number of species in CBPs is limited by the need for large populations of each species [97,102]. However, reproduction technologies and biobanks can dramatically reduce the required populations for a species in a CBPs to a few females [7,8,9,10,11,12,13,14,15]. RBCs through biobanked sperm can restore genetic variation even in highly domesticated varieties of amphibians [7,8,9,10,11,12,13,14,15]. Furthermore, the capacities of CBPs are increasing through international and regional initiatives [103,104] and potentially through the vast potential of private caregivers [25].A species’ captive care requirement should be known before establishing a CBP. However, most amphibians are amenable to captive care in simulacrums of their natural habitats or even entirely artificial habitats, as evidenced by many detailed captive care protocols and the successful reproduction of an increasing number of species [25,105,106]. If reproduction proves challenging, hormonal stimulation can assist in the reproduction of any species [8,9,11].A CBP for a species must ensure sufficient funding for its anticipated length. However, the time frame of CBPs for the species most in need cannot be predicted and could extend for years, decades, or indefinitely [107]. Many of the most significant CBPs still need to satisfy this funding requirement and have initially relied on donations and volunteers without any assurance of long-term funding [108,109].

Furthermore, the IUCN Red List [2] species recommendations as “Genome Resource Banking” [Table 2] is also not supportive of amphibian RBCs with a bias toward species of Least Concern (LC) that do not need any proactive management and with only 4.3% of EN or CE species listed as being in need of RBCs.

We provide a triage to direct resources toward species based on their endangerment status:

Triage 1. Vulnerable species. These are species with declining populations whose survival in the wild could be ensured through habitat protection or in rehabilitated, restored, or newly created habitats. If required, head-starting is the best option to produce numerous progenies for release (Table 1).

Triage 2. Endangered or Critically Endangered species with remaining habitat with a reasonable possibility of population maintenance in the wild. RBC priorities are maintaining adequate broodstock numbers to produce numerous genetically diverse progeny if needed for release (Table 1). These are the costliest RBCs because of the high broodstock numbers needed to produce progeny for release [107], especially when providing assisted gene flow [110].

Triage 3. Critically endangered species with predicted irrecoverable habitat loss. RBCs are critical, with species’ genetic diversity perpetuated through biobanked sperm. The cost of RBCs is moderate with these CBPs in institutions [13,14], and very low when conducted by private caregiver CBPs [25].

## 4. Maintaining Genetic Diversity

Maintaining a species’ genetic diversity or allelic variation in RBCs or small wild populations is expensive and inefficient without utilising biobank sperm. A minimum of 20 CBP founders captures 97.5% of heterozygosity [107]. Then, depending on the species’ generation time and lifespan, maintaining most heterozygosity for periods of only 10–25 years, without support from biobanked sperm, requires populations of 80–1500 individuals [107,111]; see Amphibian Ark calculator [112].

However, a minimum population of 40 or more founders is preferable to capture 99.5%+ of genetic diversity and increase the probability of capturing allelic variation to enable adaptability in the wild [110]. Founders can be represented by cryopreserved sperm [7,8,9,10,11,12,13,14,15,113]; however, dependent on the species fecundity, the number of females needed depends on the provision of progeny for releases (Figure 1, [114]).

An extraordinary long-term potential for amphibian RBCs is perpetuating species that would otherwise become extinct in the wild. However, RBCs can also help maintain the genetic diversity and allelic variation of species with very low populations in the wild through assisted gene flow (AGF) [87,115]. The advantages of AGF for wild populations depend on large numbers of introduced individuals providing highly advantageous alleles. However, AGF can be ineffective or even detrimental, and dependent on a wide range of factors, including natural selection, genetic drift, and outbreeding depression [116,117,118,119,120,121,122,123]. To avoid outbreeding depression with fragmented populations by AGF, the pooled genetic diversity from the core metapopulation should be provided. Isolated subpopulations should only be subject to AGF using their unique genetic diversity [107,110].

More research is needed on amphibians and other taxa to evidence the risks of AGF and the contribution of low genetic diversity or allelic variation to population declines. However, most species appear to reach extinction before genetic factors influence them [121,122], while many species have thrived with low genetic variation [21]. Furthermore, threats by major environmental stressors such as global heating have no known genetic basis to address them. Nevertheless, considering these uncertainties, ASG conducted with due diligence could benefit small fragmented or relict amphibian populations.

In conclusion, once the potential advantages of AGF to a population are evidenced, the number of released individuals at each life stage must be highly proportional to the demographic target population at that life stage. The release of large numbers of individuals in early life stages enables natural selection to maximise the potential for success. The release of individuals from CBPs to bolster population levels of small relict populations will automatically correspond to AGF [100,101,123].

## 5. Contextualising Amphibian RBCs

Current amphibian reproduction biotechnologies include stimulating reproduction and the collection of sperm or oocytes, the storage of sperm for indefinite periods, and using oocytes and sperm for artificial fertilisation. These highly advanced techniques significantly reduce the number of live individuals required for CBPs, perpetuate genetic diversity and allelic variation, and can provide numerous progenies for release [7,8,9,10,11,12,13,14,15]. Ongoing generations of anurans and salamanders have been produced from cryopreserved sperm [124,125,126]. However, the conservation crisis continues to deepen and demands the perpetuation of otherwise neglected species solely in biobanks. The satisfaction of this need should now be the focus of RBC biotechnical research [7,18,19].

Amphibian reproduction biotechnologies for conservation began through hormonal stimulation of reproductive behaviour and spawning in Russia between 1986–1989 [127,128]. Cryopreserved anuran testicular sperm was then used to fertilise oocytes at the turn of the 21st century in Russia [129,130] and Australia [131] with hormonal sperm was used in Russia in 2011 [132]. The potential of various RBCs, including cloning, to maintain amphibian genetic diversity was presented in 1999 [7]. Nonetheless, sperm-based RBCs remain the primary method for maintaining genetic diversity [133]. However, the full potential of RBCs needs to focus on developing heterocytoplasmic cloning and other somatic cell techniques [18,19]. Assisted gene flow has recently been explored through programs in the USA and Australia [87,134]. Figure 2 shows a timeline of some significant achievements of RBCs with corresponding references and other milestones in the research, and the application of RBCs, as listed in Appendix A.

## 6. Amphibian Conservation Breeding Programs [CBPs]

Amphibian conservation breeding programs (CBPs) are a significant component of RBCs that can be in range or out of range, in well-financed institutional programs such as university research groups and zoos [31,33], or be self-supporting through NGOs and their members, volunteers, ecotourism, and trade [25,30]. CBPs supported by cryopreserved sperm can guarantee species survival through a range of field species programs (Table 3, [7,8,9,10,11,12,13,14,15,135]), along with fostering conservation research, education, and community engagement [26,27,28,29,30,31,32,33].

The literature concerning CBPs uses a confusing variety of terminology. We encourage using the IUCN conventional term Conservation Breeding Programs [CBPs; Wren pers comm.]. We use the terms “in range” and “out of range” for CBPs, rather than in situ and ex situ. We provide a tabulation of a standard terminology for field programs associated with RBCs (Table 3).

Citizen conservation through private caregivers and their NGOs provides significant potential for CBPs. Private caregivers already maintain many threatened species and could support many more through community engagement, habitat protection, rehabilitation, or restoration. Increasing support for private caregivers and their NGOs and institutional RBCs focused on species in urgent need of care could help conserve all recommended species [25].

Citizen conservation turns citizens into practising conservationists: raising awareness, motivating people to get directly involved, and bringing together different areas of expertise to significantly contribute to biodiversity conservation. Citizen conservation unites private amphibian caregivers and professional conservationists in an inclusive and robust societal approach to support proactive amphibian sustainability. Zoo-based and other institutional CBPs for amphibians [8] are limited by breeding space and staff and tend to focus on charismatic or iconic species [31]. However, private caregivers (CBPs) offer almost unlimited opportunities to expand CBP capacities [25] while gaining and sharing knowledge, which is a win–win situation for the sustainable management of amphibians [140].

Greater incorporation of the vast global potential of private caregivers could lead to CBPs for threatened species at significantly reduced costs, along with building public and political support, both within and outside the species’ native range, for instance, the Responsible Herpetological Project [141]. Private caregivers have the facilities, time, passion, and knowledge to care for and reproduce threatened species (Table 4, Figure 2, [117,140,141,142]). A vast opportunity for species perpetuation exists where private caregivers’ collections as domestic varieties can later re-establish a species’ genetic diversity through biobanked sperm [7,8,9,10,11,12,13,14,15]. The success demonstrated by private caregivers in reproducing numerous species of anurans and salamanders underscores their excellent standards of care and potential to make significant contributions to CBPs if provided with the opportunity (Table 4, Figure 3, [140,141,142]).

A common misconception is the exclusion of private caregiver CBPs because of quarantine issues. Criteria for institutional CBPs are strict quarantine facilities and protocols to prevent pathogens from spreading by multiple staff working in the facility or from the external environment and biota. However, these protocols are inherent in private caregiver CBPs as they are under devoted care in an isolated domestic urban environment. Furthermore, all releases are subject to quarantine and pathogen screening, and the main pathogen threat of *Batrachochytrium* sp. has been treatable since 2012 [62]. The key to safeguarding amphibians amidst emerging infectious diseases lies in factoring in pathogens rather than attempting elimination [59,62].

Examples of the contribution of private caregiver CBPs to amphibian conservation include direct zoo collaborations, NGOs, and individual initiatives [141,142]. The amphibian conservation program at Cologne Zoo, German Republic, is a global focus and model for developing CBPs for amphibians [143]. This CBP project uses the One Plan Approach to amphibian sustainability [35], where Cologne Zoo and private caregivers work with research and habitat protection in a holistic and inclusive international program [35,140,141,142,143]. A highly respected private keeper in Germany, Karl-Heinz Jungfer, shows the ability of individuals alone to contribute. Karl-Heinz found the CE San Martin Fringe-Limbed Treefrog, *Ecnomiohyla valancifer* at a fair (Figure 3, left), a species previously known only from a handful of museum specimens, and his CBP has bred many specimens and is likely a last hope for this species. Karl-Heinz also champions a CBP for the CE demonic poison frog, *Minyobates steyermarki* (Figure 3, right).

Several amphibian genera particularly appeal to private caregiver CBPs, including the very popular *Atelopus* sp. [144], with an IUCN Red Listing of 62 CE and 14 EN species [2,144], and an AArk listing for CBPs of 30 CE and 14 EN [42], with 37 species not evaluated and several species possibly extinct in the wild. *Atelopus* sp. have established captive care protocols and offer unique opportunities for developing and applying RBCs [144]. *Neurergus* is a small but very popular salamander clade with private carers regularly breeding three VU species, but the CE *N*. *microspilotus* still needs to be included in private collections [25]. Of other salamanders, the popular warty Asian newts include thirty *Tylototriton* sp. including two CE and five EN, and eight *Paramesotriton* sp., including one CE and three EN. Fully aquatic amphibians include the Axolotl and three other CE *Ambystoma* sp., and of anurans, many highly endangered *Telmatobius* sp. [2].

## 7. Biobanking Facilities

Biobanking includes three categories: (1) “Research Biobanks”, (2) “Active Management Biobanks” for maintaining genetic diversity in CBPs or wild populations, and (3) “Perpetuity Biobanks” for species restoration [21,111,135,145]. Biobanks need ongoing financial support [38,135], with samples collected, processed, stored, and distributed within conventional guidelines and standards [135,146,147,148], and the International Society for Biological and Environmental Repositories [ISBER] promoting effective management [148].

Examples of Perpetuity Biobanks include and the Chinese Academy of Science Kunming Cell Bank [149] and Nature’s SAFE recently established in 2020 the UK [150]. Nature’s SAFE was the first cryo-network partner of the European Association of Zoos and Aquaria [EAZA] Biobank, one of Europe’s largest biobanks, collecting samples from 400 zoo and aquarium members in 48 countries. Four dedicated institutions provide qualified staff and funds for four hub biobanks that provide sample processing and facilitate sample transfer and backup [151]. The Biodiversity Biobanks South Africa project provides coordination across existing South African biodiversity biobanks. It has biobanked South Africa’s exceptionally high species diversity and endemism for over 20 years [152].

The establishment of amphibian collections in all categories of biobanks is gaining momentum [Table 5]. These include active management through the National Genome Resource Bank program at Mississippi State University, supporting RCBs toward threatened species in the USA and its territories [38]. Australia also has an active management biobank supported by universities and zoos, including sperm banking for species undergoing rapid decline [134]. Smaller projects are underway in the Global South and other developing countries [Table 5]. All these programs are supporting basic research to advance reproduction technologies. A direct link between the climate catastrophe and RBCs followed Australia’s unprecedented 2019/20 megafires [134].

However, amphibians are poorly represented in Perpetuity Biobanks. For instance, the San Diego Zoo Wildlife Alliance’s Frozen Zoo^®^ is one of the world’s largest biobanks and contains 10,000 samples [154]. The biobank houses 12.0% of EN and 14.6% of CE Mammalia, and 2.3% of EN and 4.8% of CE Reptilia/Aves. Amphibia are represented by 0.3% of EN and 0.4% of CE species, with 24 species in total, most of which are LC (not tabulated) (Table 6). The Chinese Academy of Science, Kunming Cell Bank, was established in 1986 and has now preserved 1455 cell lines from 298 animal species, including 17 amphibians [149].

## 8. Financial Considerations

### 8.1. Background

Financing for CBPs has mainly been limited to programs in wealthy Western countries, leaving a need for an independent and comprehensive provision of finance toward the biodiverse Global South and other developing countries. This need received partial recognition, with 10% of recommended finances toward CBPs in the 2007 IUCN Amphibian Conservation Action Plan. However, only 1% of finance was directed toward the critical components of RBCs, with this funding directed toward institutions in Western countries, although including their support of programs in the Global South and other developing countries (Appendix B, [155]).

The 2007 ACAP plan failed to achieve financing, but institutional initiatives have now achieved all the suggested RBC research goals ([7,8,9,10,11,12,13,14,15], Appendix A). National and international research programs have also addressed most of the listed environmental threats and their amelioration [156,157,158,159].

The revised ACAP, 2023, recognised the looming collapse of the Earth’s ecosystems and the urgent actions needed to address these challenges, and a chapter was devoted to genetic techniques, including assisted evolution. However, the chapter on CBPs did not mention private caregivers and an added chapter on amphibian RBCs mainly referred to institutions applying well-established techniques in a few projects. To prevent a predicted extinction cascade of wild species, the ACAP 2023 recommended preserving somatic cells in biobanks. However, it did emphasise the urgent need to develop restoration techniques such as cloning [153]. A similar lack of recognition of the urgent need for RBCs was shown when attendees of the 2023 Ninth World Congress of Herpetology, along with an internet survey, produced a list of research priorities that almost matched that of the ACAP 2007 [155] but with even less mention of RBCs [160].

### 8.2. Databases

Conventional databases and sample collection, processing, storage, and distribution standards are required for specimen allocation to RBCs or research and for general resource management within conservation frameworks [21,145,147,148,151,161]. Groups, including CryoArks [162] and the International Society for Biological and Environmental Repositories (ISBER, [148], promote effective management. Global agreements provide responsible and sustainable biobanking practices that balance conservation interests, equitable access, benefit sharing, research, and innovation. The United Nations Convention of Biodiversity [156], COP 15 [1], and the Nagoya Protocol for Access and Benefit-sharing [29] are international agreements for the equitable sharing of benefits from the utilisation of genetic resources and safeguarding Indigenous people’s traditional rights and stewardship of nature.

### 8.3. Costs

The costs of amphibian CBPs are highly dependent on population size, facility type, and the regional economy [12]. The largest amphibian CBPs in Western countries are well financed through collaborations between university-based research, zoos, and governmental wildlife agencies [12,13,14]. Costs in the Global South and other developing countries are considerably lower, with private caregiver CBPs being the most economical; their costs are estimated to be less than USD 5000 per species per year, to be covered by private caregivers or their organisations [25]. The use of biobanked sperm lowers costs [7,8,9,10,11,12,13,14,15,133] by a ~20-fold reduction for institutional long-term CBPs [13,14].

RBC programs in the Global South and other developing countries will rely on in-house biobanking facilities until a fully functional global network of Perpetuity Biobanks is established (Table 5). Compared to other potentials for species conservation, the costs of an in-house biobanking facility are minimal at ~US$ 15k [8,13,14]. Liquid nitrogen costs are insignificant, and cryo-vessels in Perpetuity Biobanks typically hold hundreds to thousands of samples. The estimated cost to establish amphibian cell lines in the UK is US $15,000 (Matt Guille, pers. comm), and biobanking costs per sample in the UK are USD 120 [150]. Processing fresh tissue for ten vials, including technical time, washing steps, and reagents, costs ~USD 90 per sample [150].

### 8.4. Financial Support and Management

The funding requirements of COP 15 [6], the Kunming-Montreal Global Biodiversity Framework fund [157], and the Nagoya Protocol [29] with the Global Environment Facility [26] are satisfied by amphibian RBCs. Amphibians are the most threatened terrestrial vertebrate taxon, and their eco-sociological potentials include research and education, media presence, and project development [30,31,148,160,161]. Furthermore, at a micro-economic level, local engagement includes wealth creation through native entitlement, ecotourism, and responsible trade [144,163,164,165,166,167]. RBC funding potential is also found in the Global South and other developing countries, via the trading of international debt for environmental sustainability projects [29], and in private philanthropic sources [168,169].

Nevertheless, despite these funding opportunities, securing the sustainability of biobanks takes time and effort. Most biobanks rely on public funding, while many need formal plans for long-term stewardship of their collections [170], and the recovery of costs through distribution fees often only provides partial cost recovery [171]. Financial models can address these challenges, including asymmetric pricing and advertising [172]. However, the need for sustained funding remains a critical issue, requiring the involvement of funders who understand the full funding requirements of biobanks [171,172,173,174] and emphasise the importance of considering the cost and likelihood of success in conservation projects. At the same time, Gerber [174] introduced a resource allocation framework to facilitate transparent and efficient decision-making. However, setting priorities based on the assessment of threats can be subjective and often needs more adequate data [91,92,93,94]. These and other perspectives collectively present the need for a comprehensive and transparent decision-making process toward resource allocation for amphibian RBCs [161].

## 9. Cultural Engagement

### 9.1. Biocultural Approaches and Marketing

Amphibian sustainability depends on garnering social, cultural, and political support that blends traditional, conventional, and emerging themes toward sustainability [160,165,169,175]. These extend to engagement between internationalised RBCs and traditional eco-sociological systems [176]. These partnerships include species discovery and ecological research with traditional stakeholders in highly biodiverse but under-researched regions [177] and environmental management and ecotourism with species of cultural significance [176,177]. The influence of legacy media is declining, while the internet provides a popular, accessible, and open public forum to support amphibian RBCs through engaging, welcoming, and empowering contributions [178].

Branding of amphibian RBCs requires a marketing approach with a standardised terminology and lexicon that avoids jargon [179]. Effective marketing must recognise target audience demographics and the development and fostering of brand fidelity. Simple images and emotions with social and cultural context are more effective than statistics and abstract concepts in engaging support. Solution-oriented messaging that offers direct opportunities for engagement is most rewarding [180], emphasising the role of RBCs in providing intergenerational justice for biodiversity sustainability [181,182].

### 9.2. Terraforming and Species Restoration

There are opportunities for broad social and professional engagement in amphibian RBCs through terraforming and advanced biotechnologies, including species restoration. Terraforming includes the more traditional concept of planned ecosystems in space. However, terraforming also includes the anthropogenic creation of novel ecosystems on Earth, whether planned or incidental, ongoing or historical [183]. Exciting terraforming opportunities for amphibian RBCs also include assisted evolution to provide novel genotypes with increased survivability through introducing genes, modifying existing genes, or selecting for mutations [184,185,186]. Species restoration, repopulation, or translocations can also be considered as terraforming by providing species into modified or novel ecosystems.

Amphibian RBCs could contribute to extraterrestrial terraforming through assisted evolution and the storage, transportation, and restoration of viable biomaterials [181,182,187,188]. Current space research includes amphibian fertilisation and larval development in low gravity [189,190] and a projected 200 years of sperm viability under space radiation [191]. The development of RBCs in extraterrestrial environments will reciprocally provide knowledge that contributes to Earth’s biospheric sustainability.

Biotechnical parallels between de-extinction projects and amphibian RBCs include species restoration from cryopreserved biomaterials [192]. However, unlike amphibian RBCs, de-extinction projects are challenged by low-grade donor biomaterial and surrogacy in distantly related species [192]. By surmounting these challenges, de-extinction projects will also benefit amphibian restoration through biobanked somatic cells and tissues [21]. Besides biotechnical development, de-extinction projects also engage science and futurism enthusiasts, where the vision of Colossal Biosciences has inspired institutional and private investors globally [169]. The most significant predicted contribution of de-extinction to terraforming on Earth will be to global heating amelioration through ecosystem modification of the tundra to retain methane and CO_2_ by long-extinct mammoth species [193]. Colossal Biosciences is also exemplary for seeking broad engagement with local conservation communities and agencies, and other sustainability projects, and through their internet-based publicity and outreach for biodiversity conservation [186].

### 9.3. Potentials and Pitfalls

As recently as 2022, the integration between RBCs and traditional conservation paradigms was presented as challenging [194]. This challenge has been framed as a competitive zero-sum financial game between fieldwork and RBCs, and governance using species perpetuation through RBC has been used as a reason for not supporting field programs [174]. However, regional field biologists overwhelmingly recommended RBCs during the Amphibian Arks species-needs assessments [88]. Furthermore, governments increasingly mandate “no species lost” policies that support the application of RBCs [102] and RBCs support some of the most highly financed field programs for amphibian conservation [100,101]. In any case, adopting RBCs will be expedited through improved publicity and public relations to better inform the conservation community and the public of the benefits of RBCs toward biodiversity sustainability [12,153].

The ability of amphibian RBCs alone to save many amphibian species reliably and efficiently utilising cryopreserved sperm has been possible since the turn of the millennium [194,195,196]. Subsequent research has focussed almost solely on the challenge of sperm sampling using hormonal induction rather than simply using testicular macerates, sperm storage, and producing offspring through in vitro fertilisation [133]. However, no effective initiative has established an amphibian RBC globally or developed techniques for the perpetuation of species solely in biobanks using somatic cell technologies. This lack is despite the number of amphibian species on the verge of extinction or predicted to become extinct in the wild over the next few decades [3,4,5,60].

An exemplary and cautionary narrative toward the slow development of the complete suite of amphibian RBCs necessary for biodiversity sustainability is found with corals. The development of coral RBCs was a response to mass coral diebacks. The restoration of corals from cryopreserved biomaterials was pioneered in 2006 and followed by a rapid development to application with somatic cells, tissues, and sperm by the early 2020s to support a global program for biobanking and species repopulation [197,198,199,200,201,202].

However, the field aspect of this USD 100M program failed in 2023 due to global heating killing the planted corals, leaving RBCs the best option for perpetuating these species. This example and the amphibian conservation crisis demonstrate that biobanking is needed to address the consequences of ecosystem or human societal collapses [60,189]. This realisation has encouraged conservationists to propose cosmic biobanks for coral in the frigid polar environments on the moon, at temperatures of −250 °C [203,204]. These cosmic biobanks would provide intergenerational justice toward biodiversity sustainability for the foreseeable future.

### 9.4. Global Support

Global support for amphibian RBCs includes an internet presence, provision of finance, representation at international seminars and meetings, and expert and consensual project recommendations. These activities require expertise in business management, marketing, and public relations, as well as a strong background in RBC technical and scientific expertise. The most parsimonious model to achieve these objectives is a network-based non-governmental organisation with chapters representing different technical or regional aspects of the overall project [205,206,207], including:(1)Recognition of global multi-polarity and regional, cultural, theological, and ethical traditions, i.e., support globally, act locally.(2)Democratic governance and leadership and transparent decision-making.(3)Funding through diverse funding sources to ensure long-term financial stability.(4)Focus on program spending and brand promotion to support fundraising.(5)Professional development and active social learning methods.(6)Inclusivity to ensure representation from various stakeholders, especially those in the Global South, other developing countries, and Indigenous communities.(7)Building solid collaborations and partnerships with existing conservation organisations and research institutions.(8)Deliberative processes for inclusive and informed decision-making.

## 10. The Road Ahead

Intergenerational justice and laws based on protecting the Earth and its biodiversity as legal entities provide profound cultural foundations toward biospheric sustainability. These cultural initiatives toward ecological civilisation are supported by COP 15, COP 28, and other protocols to support and finance biospheric sustainability. However, until the plateau of anthropogenic destruction of Earth’s ecosystems is reached and potentially reversed, a significant amount of biodiversity will be lost in the wild. We have demonstrated the ability of RBCs to prevent cost-effectively and reliably some of the alarming loss of amphibian diversity not protected by COPs 15 and 28.

We live in a multipolar world with increasing assertion and technical prowess in the biodiversity-rich Global South and other developing countries. However, the past has been defined by colonialism and exploitation, resulting in massive environmental destruction and cultural subjugation. Amphibian RBCs present exciting opportunities for community engagement through broad international programs, extending from local custodianship to a global presence. Including institutional and private caregivers in amphibian RBCs complements these potentials for biospheric sustainability. However, the potential for empowerment across all geopolitical and biopolitical regions still needs to be realised. Similarly, to reach their full potential, amphibian RBCs must extend beyond the research emphases on gamete collection, storage, and in vitro fertilisation, to include species perpetuation solely in biobanks through further research and the establishment of RBC facilities globally.

We provided a triage for allocation of resources to RBCs over a range of species doomed to extinction in the wild, for those species where field projects potentially ensure their survival. These approaches should be in tandem to provide the most effective use of resources from increasingly funded biospheric sustainability initiatives. Then, as corals exemplify, a dynamic multidisciplinary approach will synergise support and resource availability for amphibian sustainability in the face of the inevitable modification and loss of habitats and ecosystems.

In this era of major geopolitical realignments and environmental and social challenges, we see an opportunity for innovative strategies involving amphibian RBCs to ensure biospheric sustainability. Our vision involves establishing a globally representative organisation dedicated to championing the establishment of amphibian RBCs in the neediest regions. By promoting inclusive and democratic management through representation from the regions in need, we aim to create a robust, globally representative organisation capable of effectively advocating for RBC project development and fundraising.

These organisational requirements include expertise and the ability to form non-partisan and extensive networks with global biopolitical and geopolitical entities. Bringing together diverse voices to support a shared vision would result in a synergised global initiative that attracts expert, dynamic, and motivated contributors, especially early-career professionals, ensuring the proactive sustainability of amphibians well into the 21st century. If extended to other taxa, this model could result in further significant contributions to biospheric sustainability and intergenerational justice.

## Figures and Tables

**Figure 1 animals-14-01455-f001:**
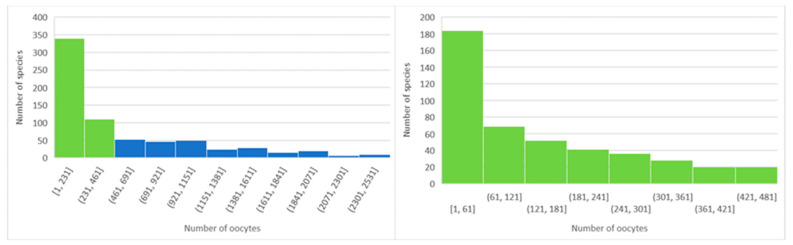
The *x*-axis shows the number of species, and the *y*-axis shows fecundity as spawn numbers. Left = the lower 90% (698 of 791 species) spawn up to 2100 oocytes, whereas the final 10% spawn up to 20,000 oocytes, mainly in Bufonid and large Ranid species. Right = the lower 50% of species mainly spawn < 70 oocytes, with many spawning < 36 oocytes, but some up 350 oocytes.

**Figure 2 animals-14-01455-f002:**
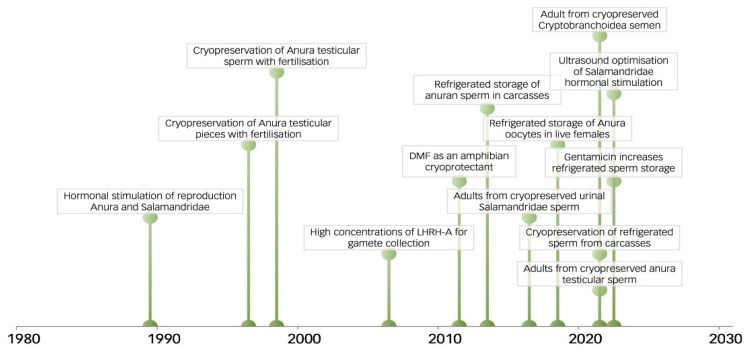
A timeline of significant milestones in developing and applying amphibian RBCs from 1980 to 2023.

**Figure 3 animals-14-01455-f003:**
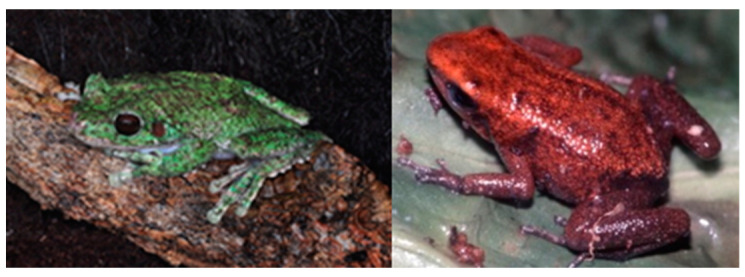
**Left** = Critically Endangered San Martin Fringe-Limbed Treefrog, *Ecnomiohyla valancifer*. **Right** = Critically Endangered demonic poison frog, *Minyobates steyermarki*. Images courtesy Peter Janzen.

**Table 1 animals-14-01455-t001:** IUCN Red List recommendations for 793 species in total for the “Ex-situ Conservation” sub-category “Captive breeding/artificial propagation” (8th August 2023 [2]). In brackets = IUCN Red List number of species in IUCN Red List Category. Left = the number recommended for “Captive breeding/artificial propagation”. Right = the % of species recommended for ‘Captive breeding/artificial propagation’. IUCN Red List Categories: NT = Near Threatened, LC = Least Concern VU = Vulnerable, EN = Endangered, CE = Critically Endangered, DD = Data Deficient can have no recommendations.

	NT	LC	VU	EN	CE	DD
Anura	34 (341) 10.0%	294 (3027) 9.7%	42 (625) 6.7%	64 (964) 6.6%	217 (591) 36.7%	209
Salamanders	0 (65) 0%	2 (189) 1.1%	5 (111) 4.5%	7 (169) 4.1%	11 (129) 8.5%	48
Caecilians	Not listed	0 (75) 0.0%	0 (4) 0.0%	0 (11) 0.0%	0 (3) 0.0%	97
Total	34 (406) 8.4%	296 (3291) 9.0%	47 (740) 6.4%	71 (1144) 6.2%	228 (723) 31.5%	na

**Table 2 animals-14-01455-t002:** The IUCN Red List recommendations for species biobanking (as Genome Resource Banking, [2]). Left = the number of species recommended for Genome Resource Banking. Centre (in brackets) = the total number of species in each IUCN Red List Category. Right = the % of species recommended for Genome Resource Banking. IUCN Red List Categories: LC = Least Concern, VU = Vulnerable, EN = Endangered, CE = Critically Endangered., DD = Data Deficient can have no recommendations.

	LC	VU	EN	CE	DD
Anura	326 (3027) 10.5%	42 (625) 6.7%	32 (964) 3.3%	23 (591) 3.9%	1000
Salamanders	13 (189) 6.9%	17 (111) 15.3%	17 (169) 10.1%	8 (129) 6.2%	48
Caecilians	0 (75) 0%	0 (4) 0%	0 (11) 0%	0 (3) 0%	97
Total	339 (3291) 10.3%	59 (740) 8.0%	49 (1144) 4.3%	31 (723) 4.3%	1145

**Table 3 animals-14-01455-t003:** Suggested conventional terms for field conservation programs.

Preferred Term	Description	Misnomers/Refs.
Species Programs		
Head Starting	The raising of individuals from eggs or embryos harvested from the wild for later release.	[136]
Repopulation	Repopulating a species in a previously populated habitat.	Reintroduction
Augmentation	The addition of captive-bred individuals to support wild populations.	Supplementation
Translocation	The movement of species, populations, or genotypes to places inside their historical range.	[136]
Relocation/Assisted Migration	The movement of species, populations, or genotypes to places outside their historical range.	[62,137]
Habitat Programs		
Mitigation	Minimising damage and maximising the eco-sustainability of environments.	[138]
Rehabilitation	The reparation of the capacity of ecosystems for biota and eco-services.	[138]
Restoration	The aspirational target of restoring ecosystems to their natural state.	[139]

**Table 4 animals-14-01455-t004:** Anurans or salamanders were surveyed in a limited canvassing of private caregiver collections in 2018 [25], providing their IUCN Red List status, number cared for/number reproduced, and percentage reproduced. CE = Critically Endangered, EN = Endangered, VU = Vulnerable, NT = Near Threatened, LC = Least Concern (from [15], Appendix A).

Endangerment	CE	EN	VU	NT	LC
Anurans	9/8, 89%	16/16, 100%	11/11, 100%	15/11, 73%	21/11, 52%
Salamanders	6/6, 100%	7/6, 86%	11/10, 91%	13/12, 92%	44/42, 95%

**Table 5 animals-14-01455-t005:** Biobanks linked to amphibian RBCs with the number of species in total, and the number of Endangered (EN) or Critically Endangered (CE) species.

Institution	Country	No. sp./IUCN List	Type	References
Taronga Conservation Society	Australia	12 sp.	sperm	[134]
University of Newcastle	Australia	26 sp.	sperm	[153]
Smithsonian Tropical Research Institute Panama	Panama	6 sp.	sperm	[153]
Centro Jambatu de Investigación y Conservación de Antibias	Ecuador	5 sp., 4 CE	sperm	Centro Jambatu de Investigación y Conservación de Anfibios
Nature’s SAFE	UK	11 sp., 4 EN or CE	sperm/repro tissue/somatic tissue	[150] R. Bolton, pers comm.
National Amphibian Genome Bank	USA	13 sp., 4 EN or CE	sperm	[38]
San Diego Wildlife Alliance and Froze Zoo	USA	26 sp., 8 EN or CE	sperm/cell lines	[154]

**Table 6 animals-14-01455-t006:** The value to the left of the brackets is the number of species biobanked in the San Diego Zoo Wildlife Alliance Frozen Zoo^®^ living cell collection (as of April 2019 [154]). The central value in brackets is the IUCN Red List species number recommendation for the Red List Category [2]. The right value is the % of species biobanked. IUCN Red List Categories: VU = Vulnerable, EN = Endangered, CE = Critically Endangered, EW = Extinct in Wild, EX = Extinct.

Class	VU	EN	CE	EW	EX
Amphibia	2 (740) 0.3%	4 (1144) 0.3%	3 (723) 0.4%	1 (2) 50%	0 (36) 0%
Reptilia/Aves	58 (1379) 4.2%	27 (1197) 2.3%	32 (666) 4.8%	3 (7) 4%	0 (191) 0%
Mammalia	80 (557) 14.4%	66 (550) 12.0%	34 (233) 14.6%	2 (2) 100%	1 (85) 1.2%
% species/% Biobanked	27.6%/1.4%	40.7%/4.1%	44.5%/4.3%	9.0%/10%	3.4%/0.3%

## Data Availability

No new data were created or analysed in this study. Data sharing is not applicable to this article.

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
