# Peer review of "Ecological Civilisation and Amphibian Sustainability through Reproduction Biotechnologies, Biobanking, and Conservation Breeding Programs (RBCs)"

_animals, 2024, doi:10.3390/ani14101455_

Round 1
Reviewer 1 Report (New Reviewer)
Comments and Suggestions for Authors
· The term RBC should be defined in the introduction - confusion on what "RBC" stands for
· In introduction, do not capitalize “endangered” and “critically endangered
· In the introduction, amphibians are cited as ideal for RBC due to their “ease of care and reproduction”; while amphibian husbandry may be simpler than that of large megafauna, proper husbandry is complex and often difficult to conduct in a fashion mimicking habitats conducive for health and reproduction; indeed, assisted reproductive technologies are often required for amphibian reproduction due to the difficulty of management. Rephrase or eliminate.
· Define “CBP” and consider explaining how the differ/relate to “RBCs”
· Typo: “We present opportunities for 1) the provision of RBCs where most needed, 2) greater integration of CBPs within RBCs, 3) significant financial support for programs in the global south and other developing countries”
· These two back-to-back lists are confusing; consider only the second list which corresponds to the paper’s sub-headings and explain items in the first list within these sub-headings: “We present opportunities for 1) the provision of RBCs where most needed, 2) greater integration of CBPs within RBCs, 3) significant financial support for programs in in the global south and other developing countries, 4) reduction of bureaucratic impediments, 5) creation of democratic, globally representative, organisation to foster and develop RBCs for amphibians and other taxa, 6) advancement of research to perpetuate species solely in biobanks [16-21]. We explore these potentials through 1) Regional Targeting of Amphibian RBCs, 2) Species Prioritisation for Amphibian RBCs, 3) Maintaining Genetic Diversity, 4) Contextualising Amphibian RBCs, 5) Amphibian Conservation Breeding Programs (CBPs), 6) Biobanking Facilities, 7) Financial Support, 8) Cultural Engagement, and 9) a Conclusion and the Road Ahead.”
· “A conclusion and the road ahead” does not seem like a necessary sub-heading – a conclusive paragraph is implied.
· Do not capitalize sub-headings listed in the introduction
· “For instance, the invasive Cane toad Rhinella marina is having major effects on some amphibian populations in Australia [68]” Consider mentioning the types of effects e.g., deleterious/competitive/predatory
· This line “To our knowledge no amphibian species have become extinct through overcollection for trade. The commercialisation of natural biological resources can yield conservation benefits through increased cultural engagement and management or harm through overexploitation” combined with the condemnation of CITES regulations reads as though the manuscript is advocating for the harvesting and transportation of wild amphibians from their endemic habitats. If this is not true, rephrase. Especially given that your next paragraph discusses
· Reduction of CITES regulations as “bureaucratic hurdles” seems extreme – you do not present enough evidence in the text to justify this inflammatory statement.
· “mythological significance to traditional communities” – please clarify your meaning here
· “Palacio et al. [94] also showed deficiencies in the Red List involving Aves hampering rather than supporting biodiversity conservation.” What is Aves hampering?
· “Even if species become domesticated in private collections” – recommend clarifying what is meant by domestication here
· “That a species captive care requirement should be known before establishing a CBP. However, there are few amphibian species that have not thrived in captivity with a plethora of published captive care protocols [25,105,106]. If reproduction is challenging RBCS can reproduce most species through hormonal stimulation [8,9,11].” This statement is wildly reductive and misleading – while several species have seen successful reproduction in captivity using hormonal stimulation, protocols remain highly variable with much species-specificity
· Figure one has no axis titles and numbers are cut off. Add titles to figure itself and expand so that values are fully visible.
Comments on the Quality of English Language
While English-language errors are not abundant, there are several instances wherein grammatical and punctuation changes should be made.
Author Response
Please see attachment

Reviewer 2 Report (New Reviewer)
Comments and Suggestions for Authors
This manuscript provides a valuable comprehensive overview of modern approaches to the conservation of amphibian biological diversity, with an emphasis on the need for cryopreservation of samples of the rarest species, the preservation of which in nature at the moment of the development of civilization presents serious difficulties. It is important that the authors' ideas are organically integrated into the current political and economic background of the development of ideas about the prospects of nature conservation, and in particular such an important taxonomic group in an ecological sense as amphibians. An important aspect of continuous financing of critical environmental technologies is discussed. I support the publication of this article on the pages of journal Animals.
However, there are a few minor comments. The presence of abbreviations in the title of the manuscript does not have clear explanations. First, the abbreviations COP15 and COP28 are not accompanied by explanations. Second, the abbreviation RBC does not correspond to the text in the title of the manuscript. It states "...Reproduction Biotechnologies, Biobanking, and Conservation Breeding Programs". Thus, it seems logical to give these abbreviations in the Abstract (with exact explenations) and in the Introduction to the manuscript (also with exact transcriptions). In the Abstract, it is necessary to decipher the abbreviations at the first mention.
The axes in Figure 1 should be clearly labeled.
The separation of “Translocation” and “Relocation” (for example, see Table 3) as separate methodological approaches is questionable. If the authors are sure that such a separation is necessary, they should provide additional arguments.
Author Response
Please see attachment

Reviewer 3 Report (New Reviewer)
Comments and Suggestions for Authors
Comments about the manuscript:
“Ecological Civilization COP 15, COP 28, and Amphibian Sustainability through Reproduction Biotechnologies, Biobanking, and Conservation Breeding Programs (RBCs)”
(Formerly : Ecological Civilization COP 15 and Amphibian Sustainability through Reproduction Technologies, Biobanking, and Conservation Breeding Programs (RBCs)
This very long manuscript (well beyond the 4000 pages recommended by the journal Animals) is a review concerning the use of new technologies to preserve amphibian cells in the long term (here red blood cells, nucleated in amphibians) in biobanks, with the aim of using them to perpetuate fragile species. Selection criteria for future uses concern several aspects: geopolitical, cultural and scientific context, biology of reproduction and development, economic aspects, etc. Genetic models are described, the potential for cryopreservation of different diploid cells to perpetuate species is taken into account, as well as the evolution of methods and techniques. The authors propose to standardize the nomenclature.
This version is a deeply second version of a previous manuscript. For my part, I find that this interesting article constitutes a useful review in the field with interesting ideas to preserve the biodiversity of species belonging to the fragile class of amphibians. The fact that it was written by several authors of different nationalities with different contexts with regard to amphibians (which belong to three orders and to very numerous species, living in temperate, humid tropical forest, Siberian zones, with different reproductive modes), certainly deserves its publication, if indeed the publisher accepts a document of some 29,000 words against the recommended 4000.
Here are some remarks.
Page 2. Write: “We argue that an a democratic globally inclusive” instead of “We argue that an a democratic globally inclusive”.
Page 2, Introduction. “potential of RBCs to satisfy”: write “RBCs” in full for the first time it appears in the text.
Page 3. Write: “3) significant financial support for programs in the” instead of “ 3) significant financial support for programs in in the”.
Page 10. “fair (Figure 3a), then known only from a handful museum specimen”: Why "3a": there is no "a" on figure 3. Use "figure 3".
Page 11. “momentum [Table 5; ibid].”: why Ibid?
Page 11, table 5: check if the reference numbers are between brackets (ex : [134], [40] , [174])
Pages 16, 17, appendix 1: write the refrence numbers are between brackets.
Author Response
Please see attachment

This manuscript is a resubmission of an earlier submission. The following is a list of the peer review reports and author responses from that submission.
Round 1
Reviewer 1 Report
Comments and Suggestions for Authors
This review addresses a key issue in our present time of mass amphibian extinctions, namely building an integrated approach to conservation breeding for these species, taking into account advances in assisted reproduction techniques and biobanking. The goal is to construct and present "the first comprehensive multidisciplinary synthesis of the state of the art and their geopolitical, cultural, and scientific backgrounds, to economically, efficiently, and reliably, perpetuate amphibian biodiversity." In my opinion, however, the article fails in reaching this ambitious goal, despite the excellent insights that can be found—sometimes—in it.
The main shortcoming is that in its present form the article is exceptionally disorganized. It is too long and consists of sections that were probably written by different people or groups of people, whose style and clarity differ considerably at times. Since the goal of the article would be to build a synthesis between the different dimensions of the problem, one might expect that the different sections (Biopolitical, Biogeographical, and Biological Background; Targeting Threats to Amphibians; Conservation Breeding Programs; Genetic Diversity and Management; RBC Technologies; Economic, Political, Cultural Factors in Biobanking) would refer conceptually and content-wise with each other. Sadly, this is mostly not the case, as the sections are in fact substantially disjointed from one another.
The whole article needs a great deal of editing work that: a) homogenizes the style; b) really builds a synthesis between what are at the present moment disjointed parts with little or no relation to each other; c) cuts out several parts and references that are inessential to the discourse, making the article more usable and thus more effective in making its case.
It is also necessary to sort through the tables, illustrations, and charts. Some boxes are unnecessary, as are many quotes placed at the beginning of paragraphs. The positioning of the tables needs to be rethought (for example, Table 1 is placed several pages after being firstly mentioned). Images should be chosen with greater care and originality or removed (the photograph of the dead frog is taken from Wikipedia).
In the current form the article not in my opinion publishable. I advise editors to reconsider the article after major revisions because the topic seems very important to me and I see some potential—even though I was initially inclined towards rejection.
Some more specific comments.
Section 2 of the article is definitely the weakest. The concept of nature is notoriously extremely complex and there is a rich literature on its meanings carried out in environmental philosophy that the authors simply ignore, opting instead to offer to the reader a definition taken from an Oxford dictionary entry. To overcome this definition taken from a dictionary entry, the authors then carve a new concept, that of "metanature," which denotes, quote, "collective phenomena of the physical world, including its components, - landscapes, plants, animals, humans, and human creations. These phenomena are recognized as interdependent, functional, and moral entities, within a Biosphere extending into space." The range of phenomena denoted in the first part of this definition is co-extensive with those already denoted by the concept of nature in its meaning of "opposite of supernatural," so the reasons for a new term are not really evident. In fact if metanature denotes the "collective phenomena of the physical world", it ends up with denoting everything that exists in the universe (and not only the "Biosphere extending into space"). The second part of the definition states that these phenomena are interdependent, functional, and moral. There are many problems with this kind of description, and I will point out only two: a) the definition claims that all things in the physical world are "functional": to whom and to what are they functional, and how can we test such an assumption?; b) the definition claims that all things in the physical world are moral: however, in what sense would a pebble or my toaster, for example, be moral? This "new" concept of metanature is introduced to overcome an extremely poor and naive definition of one of the particular meanings of nature, it is poorly defined, and it is not clear at all what purpose it can serve in the context of the objectives pursued by the article.
In many places, the authors hint to the potential of biobanking for space travel and colonization—this is an extremely interesting point. Section 2 states that the benefits of amphibian existence can be extended to space exploration and terraforming. I personally think this claim is plausible, but I cannot help but notice that the two articles cited in support of it [39 and 50] refer to biodiversity in general and do not mention amphibians explicitly.
Section 2 refers to Appendix 1, which I strongly advise the authors to cut out in its entirety. There are some misused concepts within (some examples: a theocracy is a form of government, whereas by this word the authors seem to refer in several places to organized religions; the authors write "Darwinian cultural and racial bigotry" but probably the reference they want to make is to Social Darwinism). This reviewer shares the authors' goal of building more just, equitable, and inclusive conservation strategies but also thinks that this requires a historically and sociologically rigorous approach. Many of the elements in Appendix 1 contradict each other, such as the observation that biospheric sustainability requires advanced societies with the claim that advanced countries should pursue degrowth. Moreover, many normative theses are advanced without substantive arguments to support them or data, but only through quotations—that is, through ab auctoritate.
This lack of clarity and superficiality in the definitions is also evident elsewhere in the article, such as in section 7, where it is stated that "Biocentrism presents Earth biodiversity including humanity as unique [273], meta-modernism avoids the fractures of the past and presents a holistic discourse that is both unifying and individualistically empowering [274], and metanature is an encompassing canvas on which to expand humanity's potentials within a sustainable framework (section 1). When combined, these offer a broad philosophical and practical foundation for propelling humanity toward a sustainable future.” Biocentrism and metamodernism are introduced here for the first time in the discourse and are nowhere properly defined. Their use is rather superficial. Biocentrism is not the only theory that understands biodiversity as something unique. It is not clear what "fractures of the past" meta-modernism is supposed to heal, or how this theory can present a holistic discourse that is empowering for the individual (or even what it means for an individual to be empowered). Most importantly, it is not clear why these concepts appear here out of nowhere—what is their place in the overall argument of the paper.
Similarly, again in section 7, there is a brief reference to the notion of de-extinction: "Deextinction projects conflate species conservation and animal welfare, within a global cultural and ethical framework of Biospheric Sustainability, Intergenerational Justice, and Effective Altruism". No argument or data is offered to support this statement, which is decidedly challenging from a normative point of view. Is de-extinction really a form of conservation? Are all de-extinction projects forms of conservation? Are de-extinction projects always positive from an animal welfare point of view? Why? How? Here too, there would be a vast field of literature to consider in the philosophical domain, completely ignored, in favor of an apodictic assertion whose relationship with the rest of the article is not even clear.
Reviewer 2 Report
Comments and Suggestions for Authors
I reviewed a manuscript titled “Ecological Civilization COP 15 and Amphibian Sustainability through Reproduction Technologies, Biobanking, and Conservation Breeding Programs (RBCs)” submitted to Animals. While I appreciate the authors’ attempt at using literature from diverse sources and a multidisciplinary approach to show the need for better amphibian conservation, the manuscript has too much breadth and covers too many subjects to form a cohesive narrative. As such, I cannot recommend it for publication.
The main issue is the manuscript’s focus and organization. At a macrolevel, there are too many themes/concepts and subject areas. Within the first five pages, the manuscript touches on the definition of nature itself, the ecological function of amphibians, human well-being, sustainability, amphibian diversity and biogeography, threats and disease, and conservation breeding programs without any clear direction. Not to mention, there are sections on subjects as disparate as artificial intelligence, local ecological knowledge, geopolitics, and conservation funding. It is really hard to follow.
To fix the big-picture organizational problem, I suggest reverse-outlining the entire manuscript (see https://owl.purdue.edu/owl/general_writing/the_writing_process/reverse_outlining.html). Stick to one single theme or message for the entire paper, and then find at most three big takeaways (one or two is fine, but no more than three) that you want to communicate to readers. There are enough ideas for several different papers in this single manuscript. Delete most of the paper and stick to one topic. For example, if you just reviewed evidence for amphibian RBCs, you would have enough already. Right now, there is too much going on to follow. Also, it is strange to invent a new term/acronym like "RBC" for these types of conservation initiatives that are already widely written about.
At a microlevel, there are also organizational problems throughout the manuscript. Many paragraphs are about more than one idea. For example, lines 125–136; see my comments below for other areas. After shifting the focus of the manuscript to one theme or topic, go through paragraph-by-paragraph. Ask yourself: What is the main idea? What is the purpose of this paragraph? If there is more than one idea or purpose, split each paragraph in two, delete something, and/or revise the topic sentence. Major organizational edits are needed at the paragraph level.
Certainly, I agree about the problem the manuscript seems to touch on. If traditional conservation measures were working so well, we would not be in the situation we are in. I also appreciate your perspective on amphibian conservation, which emphasizes the imbalance between where resources are for ex situ and biobanking programs and where most amphibian species occur. This view is not discussed enough in the wildlife conservation community. That said, it is not really unique to amphibians either. This might be part of the problem with the paper. If you substitute amphibians with mammals or turtles the idea is pretty much the same.
Below are some line-by-line comments. I suggest making organizational edits and cutting large sections out before addressing any of these finer-scale issues noted below.
Line 46: add “yet” or “however” before “there are no…” or maybe better change the comma to a semi-colon
Line 52: the word “is” is missing after “15”
Line 53-54: the sentence starting with “However” is not clear. I think there is some word(s) missing to complete the sentence.
Line 56: “and” is missing from before “in vitro”
Line 81: “support” should be “supporting”
Lines 94-98 and 159-162: I suggest looking for sentences that are longer than 2-3 lines throughout the manuscript and simplifying them, for example breaking them in two or removing redundant information
Lines 125-136: This paragraph seems to be about 2 different ideas. It starts off with how current conservation initiatives have neglected geographic regions with high diversity. You then write about technologies that have been neglected. Focus on one theme/idea per paragraph throughout the manuscript.
Line 139: To me, the term “amphibian biodiversity” is redundant because amphibians are alive and biological. Instead, probably “amphibian diversity” is more appropriate.
Lines 144-146: There are too many ideas in the manuscript. It reads like a wild manifesto. Focus on one area, possibly splitting this into several different manuscripts or deleting parts that do not fit in.
Line 169: capitalize only proper nouns
Figure 1: The concept of “metanature” should be defined in a separate paper, unrelated to amphibian conservation. Then, you can write about amphibian conservation as it fits within this broader concept in a different article.
Lines 220-224: I suggest putting the number of species within parentheses or otherwise making it clear the numbers refer to amphibian species richness. Also, these numbers are outdated. For example, Madagascar has over 400 described amphibian species (see amphibiaweb.org).
Line 304: Also, often conservation breeding programs within the native range of a species are still referred to as ex situ conservation because the species has been taken out of nature.
Line 322: I am not used to reading direct quotes within review articles in this field. I think it is better to just cite the article.
Lines 326-336: I think the points from 19 and 98, at least how I remember these articles, are that resources are not infinite, and species need to be prioritized for ex situ programs. You may have missed the point.
Line 334: What is Triage 1 and 3? I don’t understand this. I think the problem is you introduce the concepts before defining them. This is an organizational problem.
Lines 354-366: This is another paragraph where there are too many ideas. Stick to one main idea/theme per paragraph.
Comments on the Quality of English Language
There are grammatical and technical errors throughout, especially related to sentence structure. These should be easy to fix with a grammar-checking tool like Grammarly or even spellcheck.